# Dynamic Modeling of Land Use and Coverage Changes in the Dryland Pernambuco, Brazil

Cinthia Pereira de Oliveira [1], Robson Borges de Lima [1,*], Francisco Tarcísio Alves Junior [1],
Mayara Maria de Lima Pessoa [2], Anderson Francisco da Silva [2], Nattan Adler Tavares dos Santos [2],
Iran Jorge Corrêa Lopes [3], Cybelle Laís Souto-Maior Sales de Melo [2], Emanuel Araújo Silva [2],
José Antônio Aleixo da Silva [2] and Rinaldo Luiz Caraciolo Ferreira [2]

[1] Laboratório de Manejo Florestal, Universidade do Estado do Amapá, Macapá CEP 68901–262, Brazil;
  cinthia.oliveira@ueap.edu.br (C.P.d.O.); francisco.junior@ueap.edu.br (F.T.A.J.)
[2] Laboratório de Manejo de Florestas Naturais "José Serafim Feitosa Ferraz", Departamento de Ciência
  Florestal, Universidade Federal Rural de Pernambuco, Recife CEP 52171-900, Brazil;
  mayara.pessoa@ifmg.edu.br (M.M.d.L.P.); engf.anderson@gmail.com (A.F.d.S.);
  nattantavares@gmail.com (N.A.T.d.S.); engf.cybelle@gmail.com (C.L.S.-M.S.d.M.);
  emanuel.araujo@ufrpe.br (E.A.S.); jaaleixo@uol.com.br (J.A.A.d.S.); rinaldo.ferreira@ufrpe.br (R.L.C.F.)
[3] Departamento de Ciência Florestal, Universidade Federal do Paraná, Curitiba CEP 80210-170, Brazil;
  iranjorge._@hotmail.com
* Correspondence: robson.lima@ueap.edu.br

**Abstract:** The objective of this work was to carry out a multitemporal analysis of changes in land use and land cover in the municipality of Floresta, Pernambuco State in Brazil. Landsat images were used in the years 1985, 1989, 1993, 1997, 2001, 2005, 2009, 2014, and 2019, and the classes were broken down into areas: water, exposed soil, agriculture, and forestry, and using the Bhattacharya classifier, the classification was carried out for generating land use maps. The data was validated by the Kappa index and points collected in the field, and the projection of the dynamics of use for 2024 was constructed. The thematic maps of land use and coverage from 1985 to 2019 show more significant changes in the forest and exposed soil classes. The increase in the forest class and the consequent reduction in exposed soil are consequences of the interaction between climate and human activities and the quality of the spatial resolution of the satellite images used between the years analyzed.

**Keywords:** caatinga domain; digital classification; remote sensing

## 1. Introduction

The dry forests of the Brazilian semiarid, known as Caatinga, have been going through a continuous and lengthy reduction in their coverage [1–3]. In short, changes in land use and land cover are prominent, caused mainly by the advance in agriculture and livestock farming, raising goats and cattle, etc. [4–6], exploitation of wood and non-wood products (firewood, charcoal, fodder, etc.), in addition to urban expansion [7], as well as expansion of infrastructure and changes in the land structure. Despite being responsible for meeting the demand for forest resources in the Northeast region of Brazil, multitemporal studies on changes in their use and coverage are still incipient.

Knowledge of the human and biophysical dimensions of changes in tropical dry forests and their effects is highlighted as a priority for research [8,9]. Through remote sensing tools, there is the possibility of answers that contribute to identifying problems inherent to unrestricted land use in drylands [10] and, consequently, related to the reduction of forest cover in these areas [11].

Our current understanding of the importance of this ecosystem has been generated using remote-sensing approaches that provide spatially-explicit values relating to forest area, land cover, topography, soil, and climate variables. This information is widely used in many

dynamic models for generating predictive maps of land cover and land-use changes [4,6]. Although these maps have improved our understanding of the morphoclimatic characteristics of the caatinga, they currently do not address land-cover predictions, which are essential for environmental management.

Therefore, a better understanding of the spatial and temporal dynamics of land use forms and their potential drivers in recent years is needed to be projected into future scenarios as an effective way to inform environmental policy and decision-making. Importantly, spatially explicit scenarios can anticipate the magnitude and distribution of land-cover loss, thus providing valuable information to develop corresponding measures to manage, for example, deforestation and desertification and mitigate their impacts. Simulated scenarios can also be used to evaluate development policies, which involve proposals to build infrastructure in strategic natural systems, the establishment of land protection schemes [12], or the assessment of the combined effects of climate change (e.g., [13]).

The use of remote sensing, primarily orbital, as an aid to planning activities related to natural resources and the environment has facilitated, over the years, studies in different ecosystems [5,14–16] and allied to these techniques, spatial simulation models have been receiving greater attention from researchers, becoming a promising field of research [17,18].

Spatial or landscape simulation models simulate changes in environmental attributes across the geographic territory [19,20] and seek to help understand the causal mechanisms and development processes of environmental systems, and thereby determine how they evolve under a set of circumstances over time [21]. Therefore, data from remote sensing of the landscape and modeling together with field surveys become potentially relevant for disseminating sustainable forest management, especially in Pernambuco, as well as being essential tools for the formulation of public policies and environmental in the future region [11].

In order to provide information that better supports planning and land use in the medium term, the objective of this work was to carry out a multitemporal analysis of changes in land use and land cover in the municipality of Floresta in Pernambuco State in Brazil. As secondary objectives, we sought to (1) Understand the changes in land use and land cover from 1989 to 2019, based on the production of maps; (2) Analyze land use and land cover change (LULCC) conversions by investigating impacts resulting from 10 years (from 2014 to 2024) of changes (in a dry forest area from remote sensing tools.

## 2. Materials and Methods

### 2.1. Study Area

The study was conducted in the municipality of Floresta (Figure 1), located 433 km west of the city of Recife, in the São Francisco mesoregion and Itaparica microregion, Pernambuco, Brazil. The municipality covers an area of 3674.9 km², with an average altitude of 316 m, and is located at geographic coordinates 8°36′02″ S latitude and 38°34′05″ W longitude.

According to Köppen's climate classification, the region's climate is of the BS'h type, which reports a warm semi-arid climate. The average total annual precipitation is between 200 and 800 mm, with a concentrated rain period from January to May, with the wettest months being March and April [22]. The average annual air temperature is more significant than 26 °C. The soil in the region is classified as Chronic Luvissol, characterized as shallow and usually presenting an abrupt change in its texture [23].

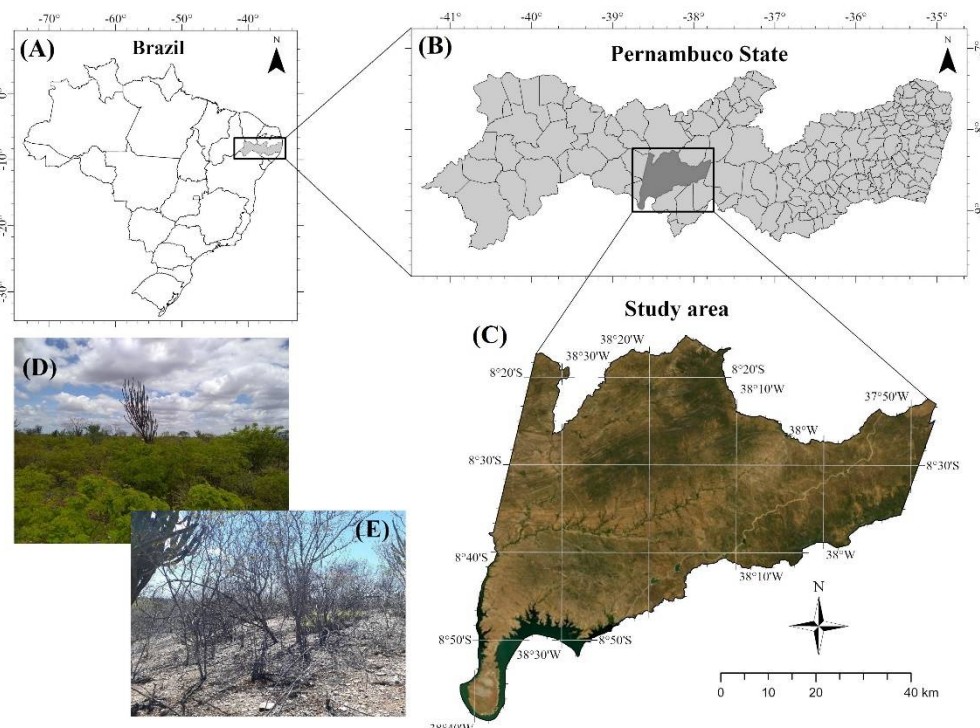

**Figure 1.** Coverage of the study area: (**A**–**C**), and photos of vegetation profiles in rainy (**D**) and dry season (**E**), in the sertão of Pernambuco, Brazil.

*2.2. Classification of Land Use and Land Cover*

Landsat-5 sensor TM (TematicMapper) satellite images from the years 1985, 1989, 1993, 1997, 2001, 2005, 2009, and Landsat-8 were used, with the sensor OLI (Operational Land Imager) from the years 2014 and 2019, acquired free of charge from the image catalog of the National Institute for Space Research (INPE), with cloud-cover rates of less than 30% and 30 m spatial resolution, for orbit/point 216/66, comprising a scene for each date evaluated; the images obtained from the TM sensor needed to be registered spatially. Image acquisition dates are shown in Table 1.

**Table 1.** Date of acquisition of the evaluated images.

| Satellite | Acquisition Date | Orbit/Point | Spatial Resolution | Spectral Bands Used | Spectral Range (μm) |
|---|---|---|---|---|---|
| Landsat 5 | 1 October 1985<br>28 October 1989<br>7 October 1993<br>2 October 1997<br>27 September 2001<br>24 October 2005<br>20 November 2009 | 216/66 | 30 | 1<br>2<br>3<br>4<br>5<br>7 | (0.45–0.52)<br>(0.52–0.60)<br>(0.63–0.69)<br>(0.76–0.90)<br>(1.55–1.75)<br>(2.08–2.35) |
| Landsat 8 | 23 March 2014<br>29 October 2019 | 216/66 | 30 | 2<br>3<br>4<br>5<br>6<br>7 | (0.45–0.51)<br>(0.53–0.59)<br>(0.64–0.67)<br>(0.85–0.88)<br>(1.57–1.65)<br>(2.11–2.29) |

For the digital classification, it was necessary to perform the image segmentation. For the Landsat-5 and Landsat-8 satellite images, the values of spectral similarity of 12 and 10 and the area sizes of 15 and 100 pixels were adopted, respectively. From the

Bhattacharya classifier implemented in the Spring software, the following thematic classes were identified:

- vegetation (areas covered with natural forest)
- farming (areas intended for agriculture and livestock)
- water (all watercourses present in the area of study)
- exposed soil (uncovered areas of vegetation and in the soil preparation phase and agricultural fallow)

The images generated from the classifications were quantified areas (hectares), according to thematic classes and generated maps of land use and land cover for all mapped dates.

To verify the reliability of the digital classification of land use and land cover in the municipality of Floresta, the Kappa index [24] was used, calculated from the confusion matrix, obtained during the training sample collection phase in each of the classified images. The acceptance intervals of the Kappa index (K) results followed the classification suggested by [24], in which it is categorized as "poor" when K is less than 0.4, "reasonable" with a K of 0,4 to 0.8, and "excellent" with K greater than 0.8.

The validation was carried out from georeferenced points in loco with a GPS device Garmin® GPSMAP 62sc (Chicado, IL, USA). A photographic record was carried out to compare the data from the digital classification of the year 2014.

*2.3. Dynamic Spatial Modeling*

For the input data of the model in the dynamic variables, only the thematic maps of land use and land cover for the years 2009 and 2014 were used, and the static variables were represented by the maps of hydrography, urban areas, road network, slope, altimetry, soils and geology of the study area. The urban area was vectored based on Landsat 5 and 8 images. The maps of the road network, water network, soils, and geology were generated from shapefiles of the State of Pernambuco from the Mineral Resources Research Company (CPRM) database. Altimetry and slope maps were generated using Spring software (version 5.2.6) based on NASA's Shuttle Radar Topography Mission (SRTM). The vector maps were converted to matrix format and standardized in the exact spatial resolution, the number of rows and columns, and the same coordinate system with Universal Transverse Mercator (UTM) and Datum WGS-84 projection.

Dynamic spatial modeling was performed in Dinamica EGO software, version 2.4.1. Moreover, it was divided into three stages: (1) construction and calibration of the model, (2) simulation, and (3) validation. The construction and calibration of the model were performed from the calculation of historical transition matrices, indicating the variation of land use and land-cover classes at two different time points, obtaining the transitions that occurred annually (multiple-step matrix) and the changes that happened in the total study interval (single-step matrix), in this case, five years, corresponding to the years between 2009 and 2014.

Once the transition rates were obtained, it was possible to perform, based on Bayes' conditional probability theorem, the method of weights of evidence adopted by Dinamica EGO for the definition of transition probabilities, which visualized the areas that are more favorable for possible changes. The procedure to follow was the calculation of the coefficients, using as input data the result of the weights of evidence method, initial and final land-use map, and static variables. The Weights of Evidence method assumes that the input maps must be spatially independent. The Cramer indices and the Join Information Uncertainty were used to assess this correlation between variables; with the selection requirement for the variables to remain in the model, a correlation threshold of 0.5 was adopted, and variables that presented a correlation above 0.5 were discarded.

The algorithms incorporated in Dinamica EGO (patcher and expander) and the isometry index and the variance of the changing area calculated in the change map were considered for the simulation model of the transitions of the spatial patterns of the use classes. In order to obtain the most suitable model, tests were carried out varying the input parameters. Model validation was performed by the fuzzy similarity comparison test

between the 2014 simulated map and the reference map for the same date; the closer to 1, the greater the similarity between the maps; thus, the distinctions being identified between the maps of actual end and initial use, and simulated ending and natural starting.

With the validation of the model, it was possible to simulate the scenarios for 2024 with the help of the SPRING software (version 5.2.6), quantifying land use and coverage and also observing the trends in class changes (Agriculture, Exposed Soil, Water, and Vegetation) on the map of initial use (2014) and of the simulated use map (2024).

## 3. Results and Discussion

### 3.1. Land Use and Coverage

The Kappa index values obtained for the municipality of Floresta using the error matrix of classified images of the years under study showed excellent acceptance levels for the most part, except for the year 2001, which was categorized as reasonable. The thematic maps obtained by the supervised digital classification process—the Bhattacharya algorithm, in the municipality of Floresta, allowed the visualization of the spatial distribution of the thematic classes (Figure 2) and their dynamic and quantification (Figure 3).

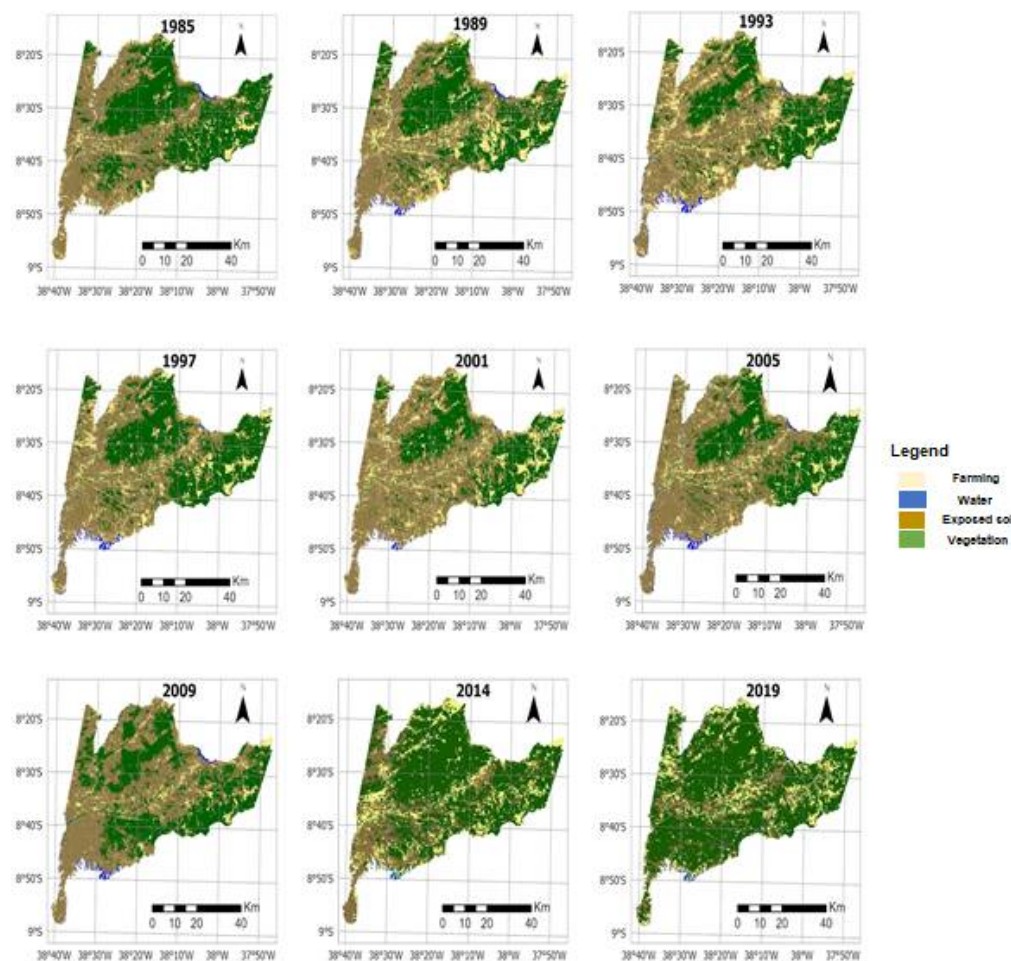

**Figure 2.** Thematic maps of land use in the municipality of Floresta-PE from 1985 to 2019.

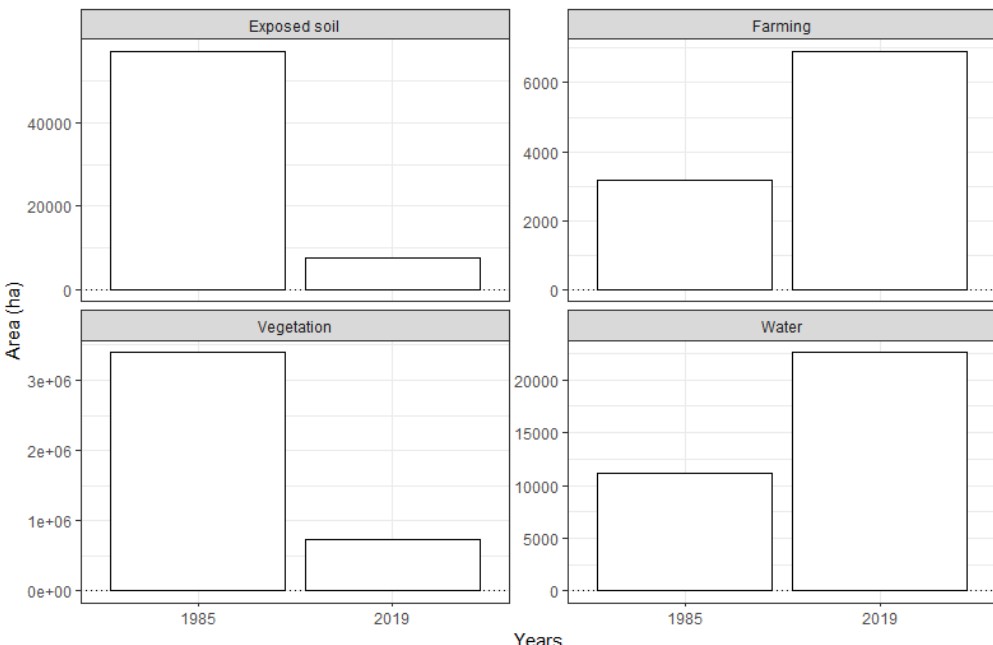

**Figure 3.** Dynamics and quantification of thematic classes through digital classification in the municipality of Floresta-PE.

Among the various areas where the 25 points marked in loco in the municipality of Floresta were collected, only seven did not correspond to the classification of the images, corresponding to 72% of correct answers. However, for the forest class, the correct answer was only 50%, which is associated with strong seasonality and high heterogeneity in terms of phytophysiognomy of the Caatinga, making the digital-image classification process difficult. Classifications in which the Kappa index indicates excellence can be found when working with a reduced number of classes [25].

The results corroborate [26] that for classifications involving four to seven classes, the use of the confusion matrix presents more minor variations. However, it is worth emphasizing the issues [27] raised regarding the basic assumptions underlying the accuracy assessment, such as generalization of the number of classes, mixed-pixel problems, incorrect category registrations, and sampling plan. It is also noteworthy that a problem associated with multitemporal remote-sensing data for detecting changes is that they do not have the same date (day/month), which varies between solar incidence angles, atmospheric conditions, and soil moisture [28].

The forest class presented an area in 2001 of 119,962.44 hectares, representing a smaller area compared to 2014 (218,602.62 ha.), equivalent to 61.70% of the area this year (Figure 3). However, for 2014 the classification was influenced by rainfall, as it was lower and unevenly distributed when observed in other periods (Figure 4). Thus, it is recommended to compare maps from 1985 to 2009, since rainfall is no more significant influence. Therefore, it can be observed that between 1985 and 2009, there was a reduction in the forest and agricultural classes, from 48.86 to 41.69% and from 10.0% to 7.58%, respectively. In comparison, the exposed-soil class increased from 40.58 to 49.64% and water from 0.58 to 1.09%.

According to Silva et al. [11], in a study in the municipality of Floresta, the removal of vegetation was necessary due to the works to transpose the São Francisco river from the east axis (started in 2007), which runs from the municipality of Floresta (PE) to Monteiro (PB), in a 220 km route in which 430 ha were deforested to make way for canals, reservoirs, construction sites, service roads and places for earth and stone extraction. Compared with the years between 1985 and 2009, the results found in this study reveal that it may be a reflection of these works, which are still in the execution phase.

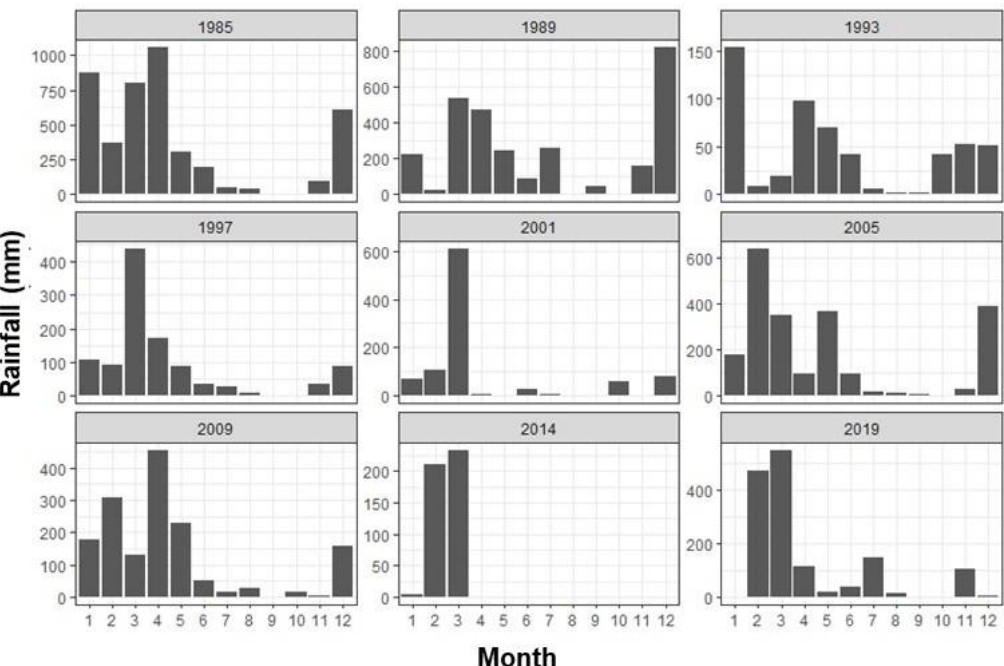

**Figure 4.** Monthly rainfall in the study area according to image acquisition dates, Floresta-PE.

The increase in exposed soil during the study period requires attention since the municipality of Floresta is inserted in the Cabrobó desertification nucleus [29,30], and according to [31], this class is a characteristic fundamental of this phenomenon in the semi-arid region of the Northeast, whose problem can worsen as a result of the successive droughts that devastate the Northeast.

There is more incredible difficulty in the digital classification of images in these areas, as the vegetation is of reduced size and greater spacing between woody individuals than in the other vegetation physiognomies of the study area, generally coinciding with the presence of the steppe savanna wooded and open. Still, the water class had a lower percentage share (0.44) in the study area, probably due to the long period of drought that has passed through this region since 2009, which did not allow the restoration of the most significant area observed in 2005 (4707.90 ha).

Except 2001 and 2014, the other years showed an increasing area of the water class, which can be explained by [2,11], due to the creation in 1988 of the Luiz Gonzaga Hydroelectric Power Plant (Itaparica) in Petrolândia-PE, which produced a greater flow of water for the Municipality of Floresta with the widening of the São Francisco River and also because this period had a more significant presence of public policies to alleviate the drought in construction of wells and weirs.

The reduction in agriculture observed between 1985 and 2009 can be explained by the fact that this class and exposed soils are closely linked, as they are part of agricultural areas [4,6,15,32]. In addition, exposed soils are generally fallow or under crop preparation [2]. Still, it may also reflect the prolonged period of drought that the region has been experiencing since 2009, corroborating the assertions of Soares [27] and Mariano et al. [31] that in times of drought, agriculture is seriously compromised. In addition, and among the main income-generating activities, the removal of firewood stands out, which, together with agriculture, promotes substantial changes in the caatinga vegetation and soils.

### 3.2. Dynamic Modeling of Land Use and Land Cover

The weight of evidence allowed us to infer what contribution a class occurred in each transition. The positive weights of evidence favor the transition's occurrence (Table 2). The highest positive values achieved in each class were considered.

**Table 2.** Continuous and static variables that most influenced land use and land cover transitions in the municipality of Floresta-PE.

| Transition | Local Variables (0 to 500 m) | Weight of Evidence (W+) |
| --- | --- | --- |
| Farming → Water | Vegetation | 0.9936 |
| Farming → exposed soil | Water | 0.759 |
| Farming → Vegetation | Water | 0.8178 |
| Water → Farming | exposed soil | 0.9434 |
| Water → exposed soil | Farming | 0.9946 |
| Water → Vegetation | Vegetation | 0.891 |
| exposed soil → Farming | Slop | 0.9445 |
| exposed soil → Water | Water | 0.9163 |
| exposed soil → Vegetation | Vegetation | 0.7112 |
| Vegetation → Farming | Farming | 0.9496 |
| Vegetation → Water | Hypsometric | 0.9581 |
| Vegetation → exposed soil | Farming | 0.9551 |

It was observed that the transition from vegetation to agriculture and exposed soil was influenced by agriculture itself, which can be explained by the high demand for this activity in the municipality. The expansion of vegetation with agriculture and exposed soil undergoing the transition to vegetation was explained by the variable water and vegetation having to be taken into account for this result when the image was obtained. The existence of dependence in the maps tested was observed only for the variable "exposed soil," in which it presented a Cramer Index (V) greater than 0.5, as for the Joint Uncertainty Index (U), this variable presented values less than 0. 5 (Table 3). As it is an essential variable for the model, it was not excluded from it.

**Table 3.** Higher Cramer Index and Joint Information Uncertainty values in the model variables.

| Variable | Cramer (V) | Uncertainty of Information Joint (U) |
| --- | --- | --- |
| Exposed soil | 0.54844644 | 0.285369436 |
| Exposed soil | 0.548146466 | 0.332466886 |
| Exposed soil | 0.547584512 | 0.367988266 |
| Exposed soil | 0.530289258 | 0.199917044 |
| Exposed soil | 0.530248771 | 0.26247358 |
| Exposed soil | 0.530062551 | 0.324499163 |

From the simulation performed in the Dinamica EGO software, the simulated map of the year 2014 was obtained, compared with the classified map of the same year to observe the quality of the model (Figure 5). The fuzzy similarity index values (Table 4), obtained from the constant and exponential decay functions for different sizes of windows with gradual clustering of pixels, presented good values in the literature.

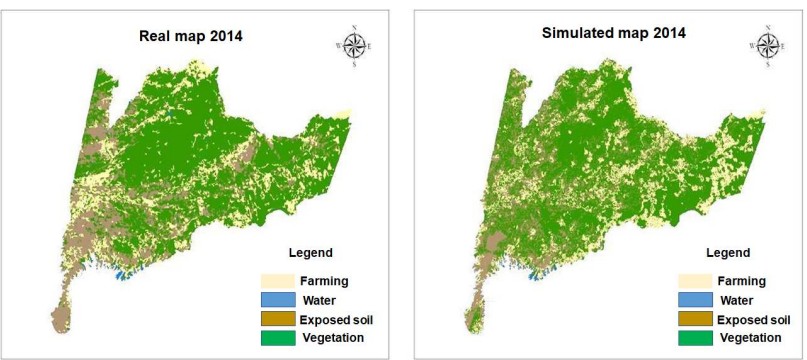

**Figure 5.** Simulated land use map for the municipality of Floresta, Pernambuco.

**Table 4.** Fuzzy similarity indices obtained from the constant and exponential decay functions for different window sizes in the periods between 2009 and 2014.

| | Similarity Index *Fuzzy* | |
|---|---|---|
| **Windows (Pixels)** | **Decay Function** | |
| | **Constant** | **Exponential** |
| 1 × 1 | 0.4713 | 0.4713 |
| 3 × 3 | 0.5944 | 0.5373 |
| 5 × 5 | 0.6448 | 0.5605 |
| 7 × 7 | 0.6820 | 0.5720 |
| 9 × 9 | 0.7132 | 0.5785 |
| 11 × 11 | 0.7399 | 0.5822 |

Ferrari [33], for example, in an Atlantic Forest ecosystem, obtained a fuzzy similarity value for 11 × 11 windows and a constant decay function of 0.84. Macedo [34], in the border region between Cerrado and Atlantic Forest, obtained a fuzzy similarity value of 0.52 as a function of constant decay for the same windows. The generation of future scenarios, or the simulation of maps a posteriori, is illustrated in Figure 6 over ten years. The first map is presented corresponding to the 2014 map used as a reference for the comparison.

Obtaining simulated maps for ten years allowed the quantification of the conversion rates of classes between the years 2014 and 2024. Table 5 shows the modeling results for the municipality of Floresta-PE.

**Table 5.** Quantification of the future scenario of the municipality of Floresta-PE and comparison with 2014.

| Class | Área 2014 (ha) | Área 2024 (ha) | 2014–2024 (ha) | 2014–2024 (%) |
|---|---|---|---|---|
| Vegetation | 218,602.62 | 229,940.64 | 11,338.02 | 5.19 |
| Farming | 55,365.75 | 61,320.78 | 5955.03 | 10.76 |
| Exposed soil | 78,790.59 | 62,569.71 | −16,220.88 | −20.59 |
| Water | 1555.47 | 801.72 | −753.75 | −48.46 |
| Total | 354,314.43 | 354,632.85 | - | - |

It was found that there is an increase linked to the areas of vegetation and agriculture in the municipality of Floresta. The areas of exposed soil had a considerable drop. According to Benedetti [25], the trend is that if the area is maintained, the same conditions as extensive activities such as agriculture will be reduced over time. From the evolutionary analysis of land use and land cover, as well as the spatial, historical survey of the occupation of the area and its prediction of how its uses will tend to behave in the future, it is possible to understand the location of the areas of these uses and the changes to which these areas are likely to be subject [19].

However, it is essential to note that our analyzes do not specifically consider direct anthropogenic factors in the modeling. Furthermore, if the effects of indirect factors, such as feedback from the surface atmosphere, were also considered, the resulting simulated vegetation area could represent significant decreases motivated by the increased severity of droughts and fires [35,36]. The risk of triggering these processes of amplification of forest loss and, therefore, reduction of vegetation cover is possibly more significant in the scenario of currently imminent climate change [37,38]. Under current deforestation trends, not only does forest loss increase, but the remaining forest areas become more fragmented, impacting their ecological functions and the future stability of the ecosystem.

Finally, while the approaches presented here help to draw relevant links between cause and effect of changing spatial points and ecological processes in tropical, dry forest landscapes, inferring complex and dynamic land-use processes is still tricky [39] because multiple processes may account for the same pattern and may change substantially because they are geographically structured [40]. To better understand the processes that drive the

observed land-cover dynamics and use [36] recommended applying dynamic models based on site-specific factors. By assessing the relative influence of different biotic and abiotic processes over longer time horizons, these models can further inform decisions about which restoration interventions will lead to spatial patterns of land use similar to those observed in reference areas. All these effects ultimately affect the ability of ecosystems to provide services to society, potentially amplifying socioeconomic inequality, which is highly documented in South America [41].

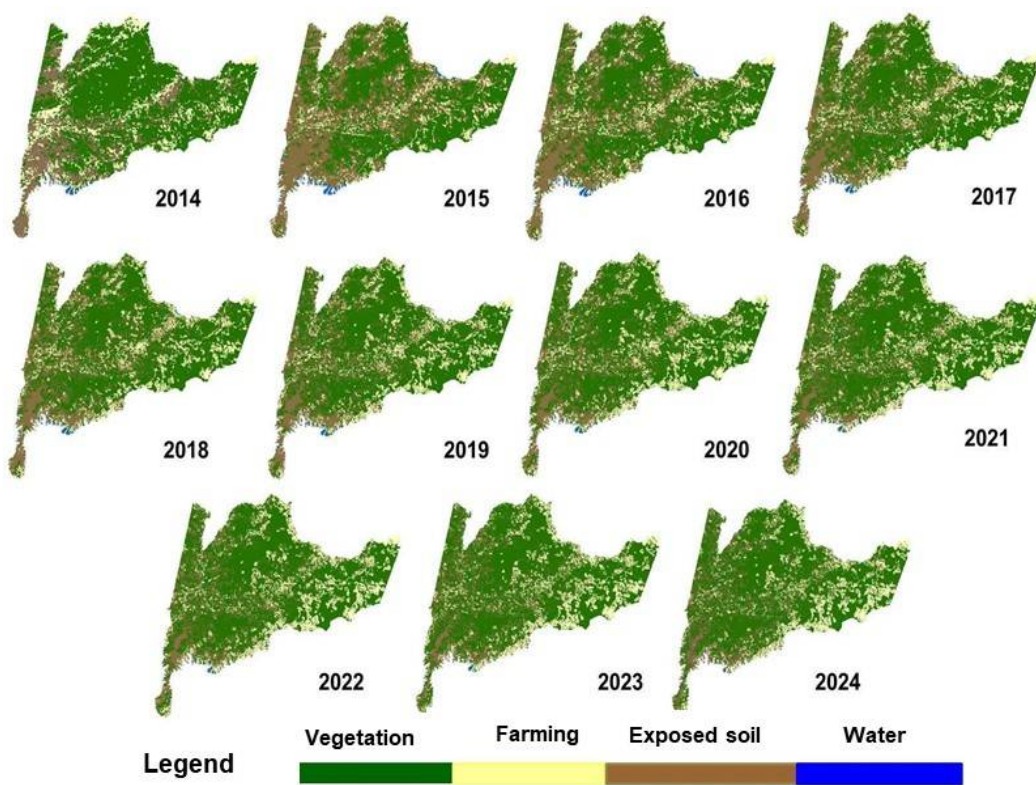

**Figure 6.** Result of the annual simulation between 2014 and 2024 for the municipality of Floresta—PE.

## 4. Conclusions

This study complements the knowledge about the direct and indirect causes of land use and land cover in tropical dry forests in Brazil. Our results indicate that from 1985 to 2014, more significant changes were observed in the forest and exposed-soil classes. The increase in forest class and the consequent reduction in exposed soil are consequences of the interaction between climate and human activities, as well as the quality of the spatial resolution of the satellite images used between the years analyzed. The low rainfall climatic conditions in the analyzed periods are primarily associated with the exposed soil throughout the municipality, as indicated by our spatially-explicit scenarios. However, their particular influences are variable in space and time and act in a complex way in combination with the other environmental drivers to produce specific trends in the transformation of the dry-forest ecosystem. These results suggest the need to complement the variables modeled in this study under the direct influence of other environmental factors inherent to the place. More specifically, our results may suggest potential future trajectories of land-cover changes, such as possible loss of vegetation area. This information is valuable for developing public policies and management strategies to combat the effects of environmental degradation and the loss of natural areas on a larger scale.

**Author Contributions:** C.P.d.O. and R.L.C.F. planned the study and wrote the manuscript, C.P.d.O., R.B.d.L., J.A.A.d.S. and M.M.d.L.P., F.T.A.J. participated in the processing data, calculation and modeling of the data and wrote the manuscript. E.A.S., A.F.d.S., N.A.T.d.S., C.L.S.-M.S.d.M., M.M.d.L.P.

and I.J.C.L. collected and processed the data images. They also commented on the manuscript. All authors have read and agreed to the published version of the manuscript.

**Funding:** This research was funded by the Coordination for the Improvement of Higher Education Personnel—Brazil (CAPES)—Financial Code 001, National Council for Scientific and Technological Development (CNPq—303991/2016-0; 400540/2016-9), Foundation for the Support of Science and Technology of the State of Pernambuco and Federal Rural University of Pernambuco (FACEPE and UFRPE—IBPG-0827-5.02/14).

**Acknowledgments:** The authors are grateful for the financial support granted by the National Council for Scientific and Technological Development (CNPq) for the productivity in research fellows of the tenth and eleventh co-authors, project leaders. The authors also thank the Universidade do Estado do Amapá for all the support in the publication of this manuscript.

**Conflicts of Interest:** The authors declare no conflict of interest.

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
