# Peer review of "Dynamic Modeling of Land Use and Coverage Changes in the Dryland Pernambuco, Brazil"

_land, doi:10.3390/land11070998_

Round 1

Reviewer 1 Report

The paper has some major flaws, even if the topic can be of interest, therefore I suggest resubmitting the paper after an extensive review. The introduction is ok, and the methodology too.

The main concern is related to the final part of the paper. There is no discussion of the results, and the conclusions are only 4 lines long. What have you learned from your research? What is the main findings you want to communicate? Limitations of the applied methodology? Future perspectives? How these results could be of interest for local planning? Discussion to compare the results with similar researches? Everything is missing.

Another relevant problem regards the 2024 land use modelling in the results section. Authors state that the maps to be compared are the ones of 1985 and 2009, as the other ones are affected by unregular rainy seasons. In the period 1985-2009 both forests and agricultural surfaces decrease. Therefore, how is it possible that 2024 land use modelling sees an increase in forests and in agricultural areas? Please clarify this issue.

other specific comments

line 70: please specify if 316 m is the average altitude, the minimum, the maximum,… in case it is the average, please, add minimum and maximum.

lines 101-102. Please add details about the forest types: what are ethe main forest types? Are they all natural? Are they managed or planned? The same for agricultural areas: what kind of agriculture is practiced? Is agroforestry practiced in the area? Is it included in forest or agricultural class?

lines 101-105. these classes (5 classes) are different from the ones applied to the maps (4 classes) in figure 2 and in figure 3. Please, use the same names and legends throughout the text.

Author Response

Dynamic Modeling of Land Use and Coverage Changes in the Dryland Pernambuco, Brazil

Oliveira et al. 2022

Author's Reply to the Review Report (Reviewer 1)

The paper has some major flaws, even if the topic can be of interest, therefore I suggest resubmitting the paper after an extensive review. The introduction is ok, and the methodology too.

The main concern is related to the final part of the paper. There is no discussion of the results, and the conclusions are only 4 lines long. What have you learned from your research? What is the main findings you want to communicate? Limitations of the applied methodology? Future perspectives? How these results could be of interest for local planning? Discussion to compare the results with similar researches? Everything is missing.

Answer: We made some substantial changes to the main body of the text, especially the final paragraphs before the conclusion. We did not separate the discussion from the results, and we did it together with a description of each main result presented.

Another relevant problem regards the 2024 land use modelling in the results section. Authors state that the maps to be compared are the ones of 1985 and 2009, as the other ones are affected by unregular rainy seasons. In the period 1985-2009 both forests and agricultural surfaces decrease. Therefore, how is it possible that 2024 land use modelling sees an increase in forests and in agricultural areas? Please clarify this issue.

Answer: For the model input data in the dynamic variables, only the thematic maps of land use and land cover from 2009 and 2014 were used. From dynamic variables of the initial (2009) and final (2014) year, the simulation model projects what can potentially happen in the following years simulated from it. The 2014 land use and land cover map directly influence the results as it was made from an image obtained in the rainy season and thus when the vegetation at its highest vigor is either in the vegetation class or in the agricultural class that includes areas of agriculture.

other specific comments

line 70: please specify if 316 m is the average altitude, the minimum, the maximum,… in case it is the average, please, add minimum and maximum.

Answer: Fixed in main text

lines 101-102. Please add details about the forest types: what are ethe main forest types? Are they all natural? Are they managed or planned? The same for agricultural areas: what kind of agriculture is practiced? Is agroforestry practiced in the area? Is it included in forest or agricultural class?

Answer: All forests identified on the map are natural. They are of the steppe savanna type with the presence of woody and sparse shrubs. For agriculture, it includes areas of monoculture and areas abandoned for fallow. There are areas with agroforestry activities, but on more minor scales, and are usually included in agriculture and farm areas.

lines 101-105. these classes (5 classes) are different from the ones applied to the maps (4 classes) in figure 2 and in figure 3. Please, use the same names and legends throughout the text.

Answer: Fixed in main text

Reviewer 2 Report

Dear Authors,

Presented paper, where the aim was was to carry out a multitemporal analysis of changes in land use and land cover in the municipality of Floresta, Pernambuco state in Braz is very importnat for scientific aspects. All stages of work were presented well. I founded some errors in grammer, so please try to improve it, ot give the text to native speaker. Please aslo check the newest literatrure, I ma thinking about years 2022-2021, which is connected with the aim of manuscript.

Author Response

Answer: We made changes to the main text to improve the grammar and understanding of the presented content

Reviewer 3 Report

This is an interesting study to analyze and simulate the land use and coverage changes in the dryland Pernambuco. This study contains loads of works and the results would be useful for local land management and interesting for readers of Land. However, there are some critical issues that the authors may need to consider:

In the introduction,

l  It should be addressed in the introduction part that how is the progress in the land cover changes research, at least in the Brazil, and why it is important to specifically study the Pernambuco, and what is the knowledge gap to fill in this research.

l  Lines 62 to 64, objective 2: compare and verify the model may not be the final purpose of your study. Simulation model is just a tool. What you want to reveal through the simulation results would be a more attractive and meaningful objective for this study.

Table 1,

l  The Acquisition Date, the formation is DD/MM/Y or MM/DD/Y?

l  Besides, it is better to explain why you choose these dates and whether they are coincidental, cause there would be different vegetation coverages in different months.

Lines 101 to 104, does this agriculture includes grassland? I noticed it enclosed livestock. If so, how to distinguish the agricultural grassland with natural grassland?

Figure 3,

l  "vegetation" or "Forest"? this issue also happened thereafter.

l  It seems the forest is intensely increased in the year 2014 and 2019, according to the figure 2.

Lines 176 to 190, we use remote sensing data to help use to discover the truth in the past year, not for guessing. Please be careful while choosing and processing the data. You can reduce some data, that is totally different with others or apparently incorrect, to improve your accuracy.

The result part is not very clear, mainly described the 1985 to 2009, and the possible reasons are not reliable.

We cannot get the conclusions from your results and discussion.

Author Response

Dynamic Modeling of Land Use and Coverage Changes in the Dryland Pernambuco, Brazil

Oliveira et al. 2022

Author's Reply to the Review Report (Reviewer 3)

This is an interesting study to analyze and simulate the land use and coverage changes in the dryland Pernambuco. This study contains loads of works and the results would be useful for local land management and interesting for readers of Land. However, there are some critical issues that the authors may need to consider:

In the introduction,

It should be addressed in the introduction part that how is the progress in the land cover changes research, at least in the Brazil, and why it is important to specifically study the Pernambuco, and what is the knowledge gap to fill in this research.

Answer: We changed the main text and sought to include this information in the introduction.

Lines 62 to 64, objective 2: compare and verify the model may not be the final purpose of your study. Simulation model is just a tool. What you want to reveal through the simulation results would be a more attractive and meaningful objective for this study.

Answer: We have changed this objective in the main text

Table 1,

The Acquisition Date, the formation is DD/MM/Y or MM/DD/Y?

Answer: We made the change to the main text by correcting it to the MM/DD/YYYY date format

Besides, it is better to explain why you choose these dates and whether they are coincidental, cause there would be different vegetation coverages in different months.

Lines 101 to 104, does this agriculture includes grassland? I noticed it enclosed livestock. If so, how to distinguish the agricultural grassland with natural grassland?

Answer: Agricultural pastures are understood as a set or type of grazing management unit, closed and separated from other areas by a fence or other barrier, and generally intended to produce forage mainly by grazing.

Natural grassland is a grassland where the original vegetation (climax vegetation) is mainly composed of herbaceous species (grasses and non-grasses) or grassland where the climax vegetation is mainly composed of herbaceous species (grasses and non-grasses) and shrubs

Figure 3,

"vegetation" or "Forest"? this issue also happened thereafter.

Answer: We made the change in the main text by correcting it to vegetation

It seems the forest is intensely increased in the year 2014 and 2019, according to the figure 2.

Lines 176 to 190, we use remote sensing data to help use to discover the truth in the past year, not for guessing. Please be careful while choosing and processing the data. You can reduce some data, that is totally different with others or apparently incorrect, to improve your accuracy.

The result part is not very clear, mainly described the 1985 to 2009, and the possible reasons are not reliable.

Answer: We made changes to the main text to improve the understanding and description of the results and conclusions.

We cannot get the conclusions from your results and discussion.

Round 2

Reviewer 1 Report

Authors have tried to improved the manuscript, especially regarding the discussion and the conclusion, but without sufficiently addressing some minor comments, as the description of what kind of agriculture is practiced (family farming, industrial farming,…) and also the use of the term “vegetation” is not appropriate. In the previous version they used forest, that can be ok, while vegetation can also include cultivated plants. Probably the term “forests and other semi-natural areas” can be a good compromise. Does exposed soil land use include urban areas?

Author Response

Kind regards again to the reviewer. We appreciate all suggestions and recommendations and hope to have answered all questions.

On the current question with regard to: whether exposed land use includes urban areas?

Our answer is that we did not include this information about urban areas, although necessary, we did not obtain all the information from the government agencies responsible at the time of the study to have greater precision in the results and improvements in the understanding of the studies on land use.

Reviewer 3 Report

The manuscript has been well-improved, but I don’t think the author has addressed and revised all of my points, for instance,

Table 1, it is better to explain why you choose these dates and whether they are coincidental, cause there would be different vegetation coverages in different months.

“Lines 101 to 104, does this agriculture includes grassland? I noticed it enclosed livestock. If so, how to distinguish the agricultural grassland with natural grassland?” for this question, the authors have explained the differences between agricultural pasture and natural grassland, however, you need to explain how you distinguish these two categories in your research with remote sensing data.

Figure 3, the authors choose to use "vegetation" to represent "Forest", and use “exposed soil” to include other vegetation, this is not professional in land use land cover research.

Figure 3, It seems the forest is intensely increased in the period 2014 to 2019, and the vegetation coverage in 2019 is much more than in 1985 according to figure 2. However, in figure 3, the vegetation in 2019 is apparently less than in 1985.

Author Response

Table 1, it is better to explain why you choose these dates and whether they are coincidental, cause there would be different vegetation coverage in different months.

Answer: The dates presented in the survey tell you about the time when we got the information from the satellite images at that time. We tried current data but unfortunately we couldn't get accurate information for the study area

“Lines 101 to 104, does this agriculture include grassland? I noticed it enclosed livestock. If so, how to distinguish the agricultural grassland with natural grassland?” for this question, the authors have explained the differences between agricultural pasture and natural grassland, however, you need to explain how you distinguish these two categories in your research with remote sensing data.

 Answer: The agricultural area does not include pastures. usually the largest area that includes agriculture is formed by local crops. Pasture information appears in very small areas and the data were not significant to assess the change in land use and cover.

Figure 3, the authors choose to use "vegetation" to represent "Forest", and use "exposed soil" to include other vegetation, this is not professional in land use land cover research.

Answer: Information on exposed soil does not refer to forest or other vegetation

Figure 3, It seems the forest is intensely increased in the period 2014 to 2019, and the vegetation coverage in 2019 is much more than in 1985 according to figure 2. However, in figure 3, the vegetation in 2019 is apparently less than in 1985 .

Answer: The period of acquisition of satellite images and also climatic factors such as increased precipitation can help to answer these differences in area with vegetation at different dates.